# DreamSmooth: Improving Model-based Reinforcement Learning via Reward Smoothing

**Vint Lee**[1]   **Pieter Abbeel**[1]   **Youngwoon Lee**[1,2]
[1]University of California, Berkeley   [2]Yonsei University

## Abstract

Model-based reinforcement learning (MBRL) has gained much attention for its ability to learn complex behaviors in a sample-efficient way: planning actions by generating imaginary trajectories with predicted rewards. Despite its success, we found that surprisingly, reward prediction is often a bottleneck of MBRL, especially for sparse rewards that are challenging (or even ambiguous) to predict. Motivated by the intuition that humans can learn from rough reward estimates, we propose a simple yet effective reward smoothing approach, *DreamSmooth*, which learns to predict a temporally-smoothed reward, instead of the exact reward at the given timestep. We empirically show that DreamSmooth achieves state-of-the-art performance on long-horizon sparse-reward tasks both in sample efficiency and final performance without losing performance on common benchmarks, such as Deepmind Control Suite and Atari benchmarks.

## 1 Introduction

Humans often plan actions with a rough estimate of future rewards, instead of the exact reward at the exact moment (Fiorillo et al., 2008; Klein-Flügge et al., 2011). A rough reward estimate is mostly sufficient to learn a task, and predicting the exact reward is often challenging since it can be ambiguous, delayed, or not observable. Consider for instance the manipulation task illustrated in Figure 1 (middle) of pushing a block on a table into a bin, where a large reward is given only on the timestep when the block first touches the bin. Using the same image observations as the agent, it is challenging even for humans to predict the correct sequence of rewards. Crucially, this issue is present in many environments, where states with no reward are almost indistinguishable from those with rewards.

An accurate reward model is vital to model-based reinforcement learning (MBRL) – reward estimates that are too high will cause an agent to choose actions that perform poorly in reality, and estimates that are too low will lead an agent to ignore high re-

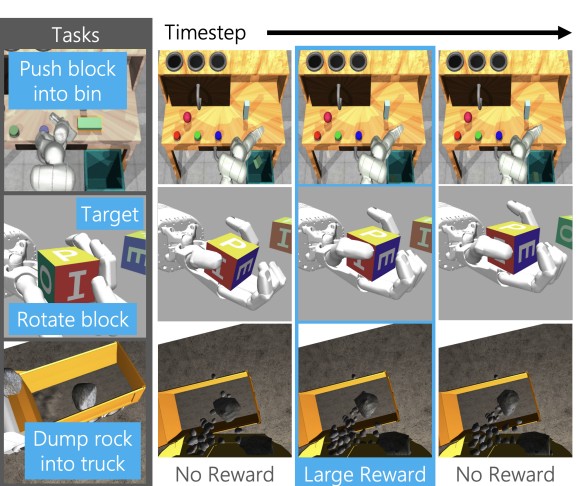

Figure 1: Predicting the exact sequence of rewards is extremely difficult. These examples show the sequences of image observations seen by the agent just before and after it receives a large reward. There is little to visually distinguish timesteps with a large reward from those without, which creates a significant challenge for reward prediction.

wards. Despite its difficulty and importance, the reward prediction problem in MBRL has been largely overlooked. We find that even for the state-of-the-art MBRL algorithm, DreamerV3 (Hafner et al., 2023), reward prediction is not only challenging, but is also a performance bottleneck for many tasks. For instance, DreamerV3 fails to predict any reward for most objectives in the Crafter environment (Hafner, 2022) with similar failure modes observed on variants of the RoboDesk (Kannan et al., 2021) and Shadow Hand (Plappert et al., 2018) tasks.

Inspired by the human intuition that only a rough estimate of rewards is sufficient, we propose a simple yet effective solution, **DreamSmooth**, which learns to predict a temporally-smoothed reward rather than the exact reward at each timestep. This makes reward prediction much easier – instead of having to predict rewards exactly, now the model only needs to produce an estimate of when large rewards are obtained, which is sufficient for policy learning.

Our experiments demonstrate that while extremely simple, this technique significantly improves performance of different MBRL algorithms on many sparse-reward environments. Specifically, we find that for DreamerV3 (Hafner et al., 2023), TD-MPC (Hansen et al., 2022), and MBPO (Janner et al., 2019), our technique is especially beneficial in environments with the following characteristics: sparse rewards, partial observability, and stochastic rewards. Finally, we show that even on benchmarks where reward prediction is not a significant issue, DreamSmooth does not degrade performance, which indicates that our technique can be universally applied.

## 2 RELATED WORK

Model-based reinforcement learning (MBRL) leverages a dynamics model (i.e. world model) of an environment and a reward model of a desired task to plan a sequence of actions that maximize the total reward. The dynamics model predicts the future state of the environment after taking a specific action and the reward model predicts the reward corresponding to the state-action transition. With the dynamics and reward models, an agent can simulate a large number of candidate behaviors in imagination instead of in the physical environment, allowing MBRL to tackle many challenging tasks (Silver et al., 2016; 2017; 2018).

Instead of relying on the given dynamics and reward models, recent advances in MBRL have enabled learning a world model of high-dimensional observations and complex dynamics (Ha & Schmidhuber, 2018; Schrittwieser et al., 2020; Hafner et al., 2019; 2021; 2023; Hansen et al., 2022), as well as a temporally-extended world model (Shi et al., 2022). Specifically, DreamerV3 (Hafner et al., 2023) has achieved the state-of-the-art performance across diverse domains of problems, e.g., both with pixel and state observations as well as both with discrete and continuous actions.

For realistic imagination, MBRL requires an accurate world model. There have been significant efforts in learning better world models by leveraging human videos (Mendonca et al., 2023), by adopting a more performant architecture (Deng et al., 2023), and via representation learning, such as prototype-based (Deng et al., 2022) and object-centric (Singh et al., 2021) state representations, contrastive learning (Okada & Taniguchi, 2021), and masked auto-encoding (Seo et al., 2022; 2023).

However, compared to the efforts on learning a better world model, learning an accurate reward model has been largely overlooked. Babaeizadeh et al. (2020) investigates the effects of various world model designs and shows that reward prediction is strongly correlated to task performance when trained on an offline dataset, while limited to dense-reward environments. In this paper, we point out that accurate reward prediction is crucial for MBRL, especially in sparse-reward tasks and partially observable environments, and propose a simple method to improve reward prediction in MBRL.

## 3 APPROACH

The main goal of this paper is to understand how challenging reward prediction is in model-based reinforcement learning (MBRL) and propose a simple yet effective solution, *reward smoothing*, which makes reward prediction easier to learn. In this section, we first provide a background about MBRL in Section 3.1, then present experiments demonstrating the challenge of predicting sparse reward signals in Section 3.2, and finally explain our approach, DreamSmooth, in Section 3.4.

### 3.1 BACKGROUND

We formulate a problem as a partially observable Markov decision process (POMDP), which is defined as tuple $(\mathcal{O}, \mathcal{A}, P, R, \gamma)$. $\mathcal{O}$ is an observation space, $\mathcal{A}$ is an action space, $P(\boldsymbol{o}_{t+1}|\boldsymbol{o}_{\leq t}, \boldsymbol{a}_{\leq t})$ with timestep $t$ is a transition dynamics, $R$ is a reward function that maps previous observations and actions to a reward $r_t = R(\boldsymbol{o}_{\leq t}, \boldsymbol{a}_{\leq t})$, and $\gamma \in [0, 1)$ is a discount factor (Sutton & Barto, 2018). RL aims to find a policy $\pi(\boldsymbol{a}_t \,|\, \boldsymbol{o}_{\leq t}, \boldsymbol{a}_{<t})$ that maximizes the expected sum of rewards $\mathbb{E}_\pi[\sum_{t=1}^{T} \gamma^{t-1} r_t]$.

This paper focuses on MBRL algorithms that learn a world model $P_\theta(z_{t+1}|z_t, a_t)$ and reward model $R_\theta(r_t|z_t)$ from agent experience, where $z_t$ is a learned latent state at timestep $t$. The learned world model and reward model can then generate imaginary rollouts $\{z_\tau, a_\tau, r_\tau\}_{\tau=t}^{t+H-1}$ of the horizon $H$ starting from any $z_t$, which can be used for planning (Argenson & Dulac-Arnold, 2021; Hansen et al., 2022) or policy optimization (Ha & Schmidhuber, 2018; Hafner et al., 2019). Specifically, we use the state-of-the-art algorithms, DreamerV3 (Hafner et al., 2023) and TD-MPC (Hansen et al., 2022).

DreamerV3 (Hafner et al., 2023) uses the predicted rewards for computing new value targets to train the critic. For learning a good policy, the reward model plays a vital role since the critic, from which the actor learns a policy, receives its training signal exclusively through the reward model. Note that the data collected from the environment is only used for training a world model and reward model.

On the other hand, TD-MPC (Hansen et al., 2022) learns a state-action value function $Q(z_t, a_t)$ directly from agent experience, not from predicted rewards. However, the reward model is still important for obtaining a good policy in TD-MPC because the algorithm uses both the reward model and value function to obtain the policy through online planning.

### 3.2 REWARD PREDICTION IS DIFFICULT

Reward prediction is surprisingly challenging in many environments. Figure 1 shows sequences of frames right before and after sparse rewards are received in diverse environments. Even for humans, it is difficult to determine the exact timestep when the reward is received in all three environments.

We hypothesize that the mean squared error loss $\mathbb{E}_{(z,r)\sim\mathcal{D}}[(R_\theta(z) - r)^2]$, typically used for reward model training, deteriorates reward prediction accuracy when there exist *sparse rewards*. This is because predicting a sparse reward a single step earlier or later results in a higher loss than simply predicting $0$ reward at every step. Thus, instead of trying to predict sparse rewards at the exact timesteps, a reward model minimizes the loss by entirely omitting sparse rewards from its predictions.

To verify this hypothesis, we plot the ground-truth and DreamerV3's predicted rewards in Figure 2. On the four tasks described in Section 4.1, the reward models struggle at predicting exact rewards and simply ignore sparse rewards unless they are straightforward to predict. This hypothesis also holds in a deterministic and fully-observable environment, Crafter, which has 24 sources of sparse rewards. The reward model fails to predict most of these reward sources (Figure 2d).

The difficulty of reward prediction can be further exacerbated by partial observability, ambiguous rewards, or stochastic dynamics of environments. As an example in the first (third) row in Figure 1,

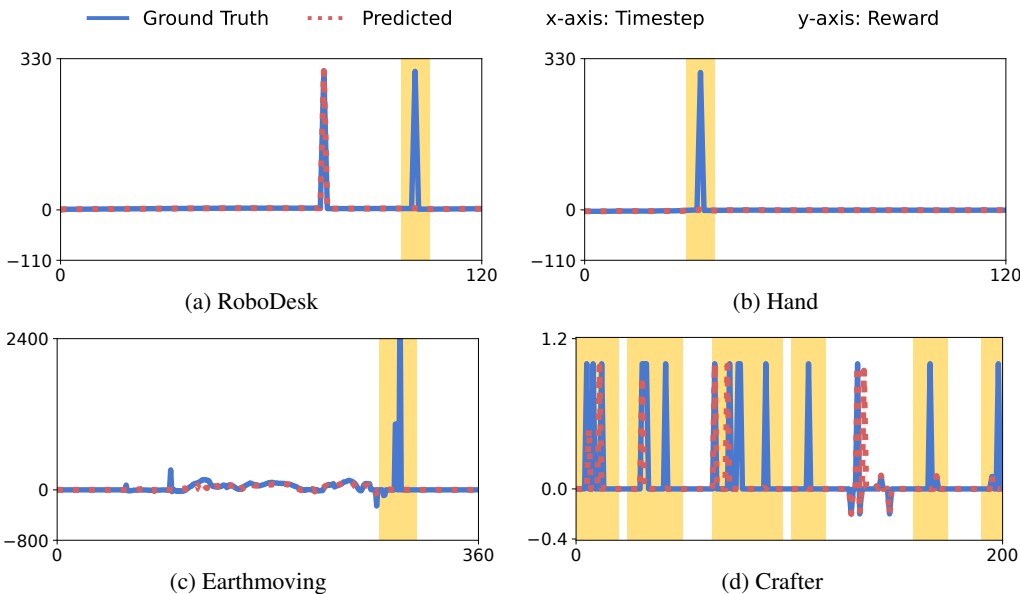

Figure 2: Ground truth rewards and DreamerV3's predicted rewards over an evaluation episode. The reward model misses many *sparse* rewards, which is highlighted in yellow.

the sparse rewards are given when the block (the rocks in the third example) first contacts the bin (the dumptruck). The exact moment of contact is not directly observable from the camera viewpoint, and this makes reward prediction ambiguous. Moreover, stochastic environment dynamics, e.g., contact between multiple rocks, can make predicting a future state and reward challenging.

### 3.3 REWARD PREDICTION IS A BOTTLENECK OF MBRL

The preceding section shows that reward prediction is challenging in many environments. More importantly, this poor reward prediction can be a bottleneck of policy learning, as shown in Figure 3. In RoboDesk, where the reward model does not reliably detect the completion of the second task (Figure 2a), the policy gets stuck at solving the first task and fails on subsequent tasks. In Earthmoving, where the reward model cannot capture rewards for successful dumping (Figure 2c), the policy frequently drops the rocks outside the dumptruck. These consistent failure modes in reward prediction and policy learning in DreamerV3 suggest that poor reward prediction can be a bottleneck of MBRL.

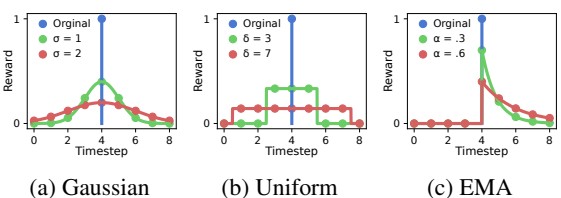

(a) RoboDesk (b) Earthmoving

Figure 3: The reward model's inability to predict sparse rewards for completing tasks leads to poor task performance. (a) In RoboDesk, the agent gets stuck after learning the first task, and is unable to learn to perform the subsequent tasks. (b) In Earthmoving, the policy often fails to dump the rocks accurately into the dumptruck. The learning curves are averaged over 3 seeds.

### 3.4 DREAMSMOOTH: IMPROVING MBRL VIA REWARD SMOOTHING

To address the reward prediction problem, we propose a simple yet effective solution, **DreamSmooth**, which relaxes the requirement for the model to predict sparse rewards at the exact timesteps by performing temporal smoothing. Allowing the reward model to predict rewards that are off from the ground truth by a few timesteps makes learning easier, especially when rewards are ambiguous or sparse.

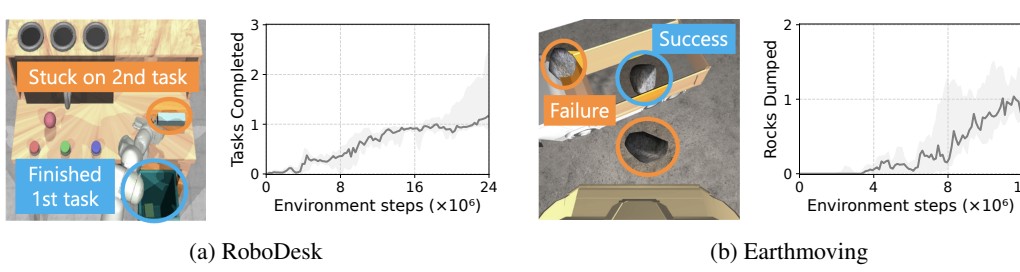

(a) Gaussian (b) Uniform (c) EMA

Figure 4: Reward smoothing on sparse reward 1 at $t = 4$. $\sigma$, $\delta$, and $\alpha$ are smoothing hyperparameters.

Specifically, DreamSmooth applies temporal smoothing to the rewards upon collecting each new episode. DreamSmooth can work with any smoothing function $f$ that preserves the sum of rewards:

$$\tilde{r}_t \leftarrow f(r_{t-L:t+L}) = \sum_{i=-L}^{L} f_i \cdot r_{\text{clip}(t+i,0,T)} \quad s.t. \quad \sum_{i=-L}^{L} f_i = 1, \quad (1)$$

where $T$ and $L$ denote the episode and smoothing horizons, respectively. For simplicity, we omit the discount factor in Equation (1); the full equation can be found in Appendix, Equation (6). Episodes with the smoothed rewards are stored in the replay buffer and used to train the reward model. The agent learns only from the smoothed rewards, without ever seeing the original rewards. The smoothed rewards ease reward prediction by allowing the model to predict rewards several timesteps earlier or later, without incurring large losses. In this paper, we investigate three popular smoothing functions: Gaussian, uniform, and exponential moving average (EMA) smoothing, as illustrated in Figure 4.

While the main motivation for smoothing is to make it easier to learn reward models, we note that reward smoothing in some cases preserves optimality – *an optimal policy under smoothed rewards $\tilde{r}$ is also optimal under the original rewards $r$*. In particular, we provide a proof in Appendix A for the

optimality of EMA smoothing (and any smoothing function where $\forall i > 0, f_i = 0$) by augmenting the POMDP states with the history of past states. However, when future rewards are used for smoothing (e.g. Gaussian smoothing), the smoothed rewards are conditioned on policy, and we can no longer define an equivalent POMDP. In such cases, there is no theoretical guarantee. Even so, we empirically show that reward models can adapt their predictions alongside the changing policy, and achieve performance improvements.

The implementation of DreamSmooth is extremely simple, requiring only one additional line of code to existing MBRL algorithms, as shown in Algorithm 1. The overhead of reward smoothing is minimal, with time complexity $O(T \cdot L)$. More implementation details can be found in Appendix B.

---

**Algorithm 1** COLLECT_ROLLOUT ($\pi$: policy, $\mathcal{D}$: replay buffer) in DREAMSMOOTH

$\{(\boldsymbol{o}_t, \boldsymbol{a}_t, r_t)_{t=1}^T\} \leftarrow \text{ROLLOUT}(\pi)$
$\{r_t\}_{t=1}^T \leftarrow \text{GAUSSIAN}(\{r_t\}_{t=1}^T, \sigma) \text{ or } \text{EMA}(\{r_t\}_{t=1}^T, \alpha)$     ▷ **only one** line needs to be added.
$\mathcal{D} \leftarrow \mathcal{D} \cup \{(\boldsymbol{o}_t, \boldsymbol{a}_t, r_t)_{t=1}^T\}$

---

## 4 EXPERIMENTS

In this paper, we propose a simple reward smoothing method, DreamSmooth, which facilitates reward prediction in model-based reinforcement learning (MBRL) and thus, improves the performance of existing MBRL methods. Through our experiments, we aim to answer the following questions: (1) Does reward smoothing improve reward prediction? (2) Does better reward prediction with reward smoothing lead to better sample efficiency and asymptotic performance of MBRL in sparse-reward tasks? (3) Does MBRL with reward smoothing also work in common dense-reward tasks?

### 4.1 TASKS

We evaluate DreamSmooth on four tasks with sparse subtask completion rewards and two common RL benchmarks. Earthmoving uses two $64 \times 64$ images as an observation while all other tasks use a single image. See Appendix C for environment details.

- **RoboDesk:** We use a modified version of RoboDesk (Kannan et al., 2021), where a sequence of manipulation tasks (`flat_block_in_bin`, `upright_block_off_table`, `push_green`) need to be completed in order (Figure 5a). We use the original dense rewards together with a large sparse reward for each task completed.
- **Hand:** The Hand task (Plappert et al., 2018) requires a Shadow Hand to rotate a block in hand into a specific orientation. We extend it to achieve a sequence of pre-defined goal orientations in order. In addition to the original dense rewards, we provide a large sparse reward for each goal.
- **Earthmoving:** The agent controls a wheel loader to pick up rocks and dump them in the dump truck (Figure 5c). Sparse rewards are given for picking up and dumping rocks, with dense rewards for moving rocks towards the dump truck. The environment is simulated using the AGX Dynamics physics engine (Algoryx, 2020) with the AGX Terrain module (Servin et al., 2021).
- **Crafter:** Crafter (Hafner, 2022) is a minecraft-like 2D environment, where the agent tries to collect, place, and craft items in order to survive. There are 22 achievements in the environment

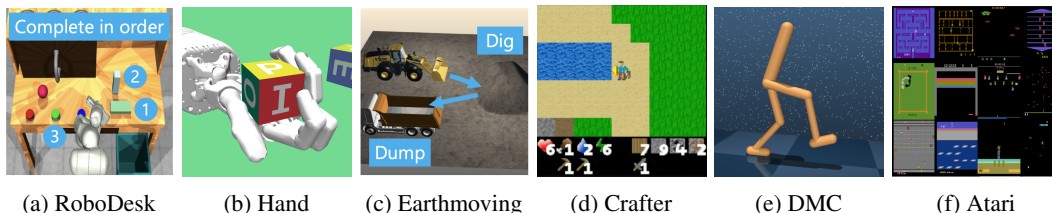

(a) RoboDesk     (b) Hand     (c) Earthmoving     (d) Crafter     (e) DMC     (f) Atari

Figure 5: We evaluate DreamSmooth on four tasks with sparse subtask completion rewards (a-d). We also test on two popular benchmarks, (e) DeepMind Control Suite and (f) Atari.

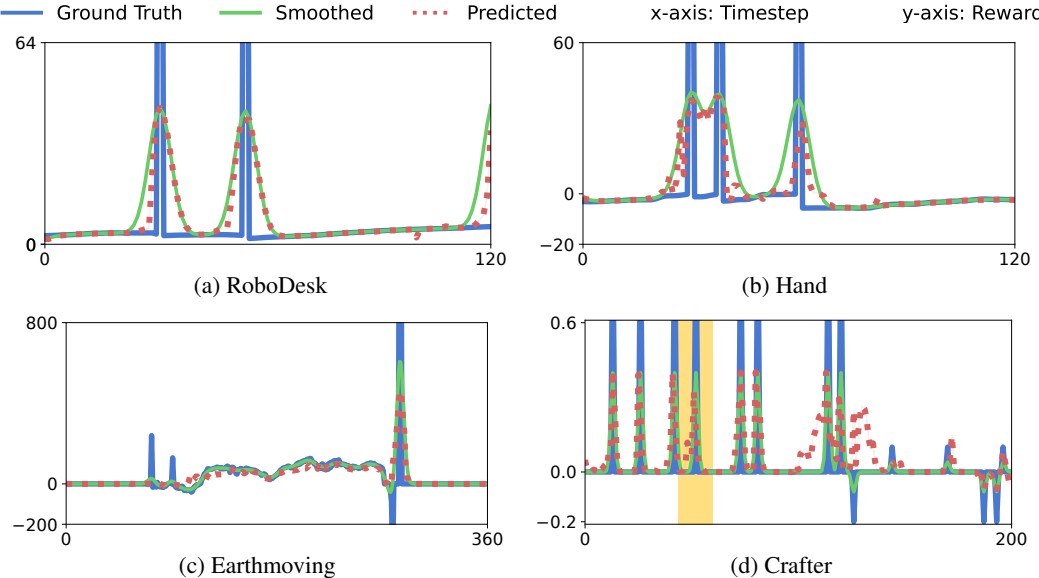

Figure 6: We visualize the ground truth rewards, smoothed rewards with Gaussian smoothing, and predicted rewards by DreamerV3 trained on the smoothed rewards over an evaluation episode. In contrast to Figure 2, the reward models with reward smoothing capture most of sparse rewards.

(e.g. collecting water, mining diamonds) with a sparse reward 1 for obtaining each achievement for the first time. A small reward is given (or lost) for each health point gained (or lost).

- **DMC:** We benchmark 7 DeepMind Control Suite continuous control tasks (Tassa et al., 2018).
- **Atari:** We benchmark 6 Atari tasks (Bellemare et al., 2013) at 100K steps.

## 4.2 IMPROVED REWARD PREDICTION WITH REWARD SMOOTHING

We first visualize the ground truth rewards, smoothed rewards (Gaussian smoothing), and reward prediction results of DreamerV3 trained with DreamSmooth in Figure 6. We observe that reward smoothing leads to a significant improvement in reward prediction: DreamSmooth successfully predicts most of the (smoothed) sparse rewards and no longer omits vital signals for policy learning or planning.

The improvement is especially notable in Crafter. In Figure 7, we measure the accuracy of the reward model, (i.e. predicting a reward larger than half of the original or smoothed reward for DreamerV3 and DreamSmooth respectively) at the exact timesteps for each subtask. The vanilla DreamerV3's reward model (baseline) misses most of the sparse rewards while DreamSmooth predicts sparse rewards more accurately in 15/19 subtasks.

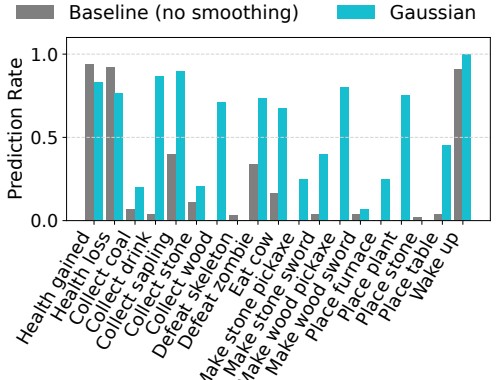

Figure 7: Reward prediction rates for 19 achievements in Crafter. The other 3 tasks have been never achieved by both methods. With reward smoothing, the prediction rates are better in 15/19 tasks.

## 4.3 RESULTS

We compare the vanilla DreamerV3 (Hafner et al., 2023) with DreamSmooth, whose backbone is also DreamerV3. For DreamSmooth, we evaluate Gaussian, uniform, and EMA smoothing. The hyperparameters for DreamerV3 and smoothing functions can be found in Appendix B. As shown in Figure 8, DreamSmooth-Gaussian and DreamSmooth-Uniform significantly improve the performance

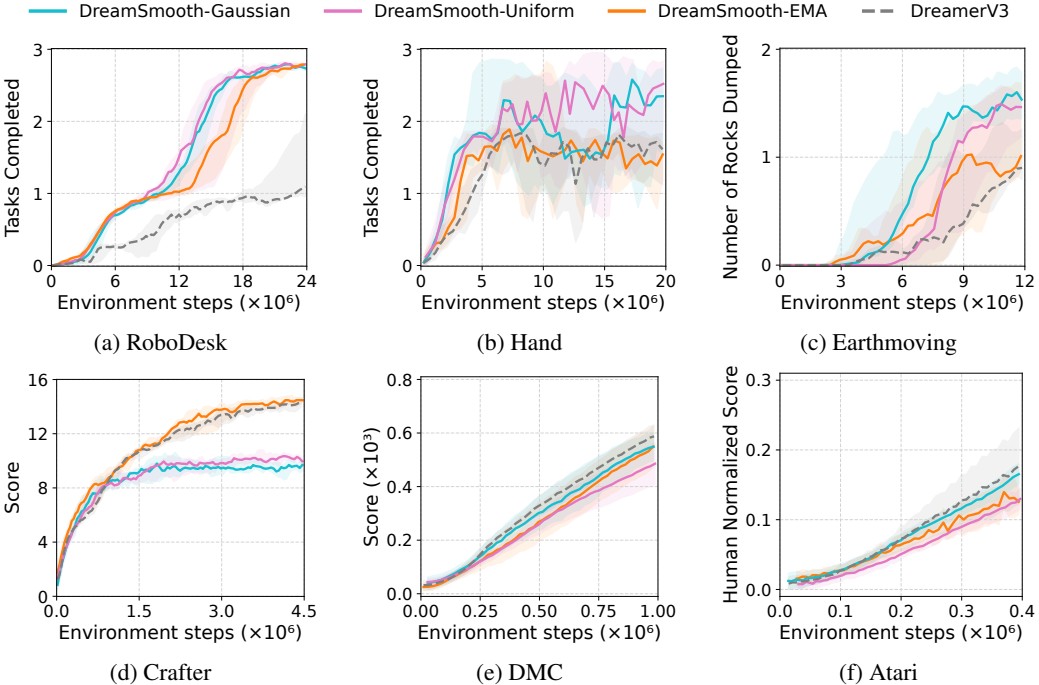

Figure 8: Comparison of learning curves of DreamSmooth (Gaussian, Uniform, EMA) and DreamerV3. The shaded regions in (a-d) show the maximum and minimum over 3 seeds. For DMC (e) and Atari (f), we aggregate results over 7 and 6 tasks respectively, and display the standard deviation.

as well as the sample efficiency of DreamerV3 on the Robodesk, Hand, and Earthmoving tasks. The only change between DreamerV3 and ours is the improved reward prediction, as shown in Section 4.2. This result suggests that *reward prediction is one of major bottlenecks of the MBRL performance*.

While all smoothing methods lead to improvements over DreamerV3, Gaussian smoothing generally performs the best, except on Crafter, with uniform smoothing showing comparable performance. The better performance of Gaussian and uniform smoothing could be because it allows predicting rewards both earlier and later, whereas EMA smoothing only allows predicting rewards later.

Despite the improved reward prediction accuracy, DreamSmooth-Gaussian and DreamSmooth-Uniform perform worse than the baseline in Crafter. This could be because the symmetric Gaussian and Uniform smoothing kernels require the reward models to anticipate future rewards, while EMA smoothing does not. We believe this leads to more false-positive reward predictions from the former, leading to poor policy learning in Crafter. More details can be found in Appendix F.

We also observe that on the DMC and Atari benchmarks, where reward prediction is not particularly challenging, our technique shows comparable performance with the unmodified algorithms (see Appendix, Figure 17 for full results), suggesting that reward smoothing can be applied generally.

In Figure 9, DreamSmooth also improves the performance of TD-MPC (Hansen et al., 2022) and MBPO (Janner et al., 2019). In the Hand task, the vanilla algorithms are unable to consistently solve the first task, even with proprioceptive state observations. However, DreamSmooth enables both algorithms to complete the tasks, even learning on pixel observations with TD-MPC. This suggests that DreamSmooth can be useful in a broad range of MBRL algorithms that use a reward model. We only demonstrate the

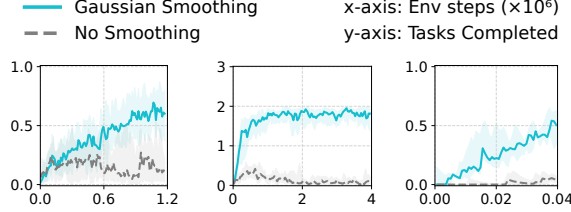

(a) TD-MPC (Pixel) (b) TD-MPC (State) (c) MBPO (State)

Figure 9: Learning curves (median over 3 seeds) for TD-MPC and MBPO, with and without DreamSmooth, on the Hand task.

Hand task since TD-MPC and MBPO fail on other sparse-reward tasks, with MBPO requiring demonstrations to make progress on the Hand task (see Appendix E for details).

## 4.4 ABLATION STUDIES

**Data Imbalance.** One possible cause of poor reward predictions is data imbalance – the reward model trains on few examples of sparse rewards due to their infrequency, potentially leading to poor predictions. To test this hypothesis, we conducted experiments with oversampling: with probability $0.5$, we sample a sequence containing sparse rewards; otherwise, we sample uniformly from all sequences in the buffer. As shown in Figure 10, oversampling performs better than the baseline, but learns slower than DreamSmooth. This suggests that while data imbalance contributes to the difficulty of reward prediction, it is not the only factor hindering performance. Furthermore, this oversampling method requires domain knowledge about which reward signals are to be oversampled while DreamSmooth is agnostic to the scale and frequency of sparse rewards.

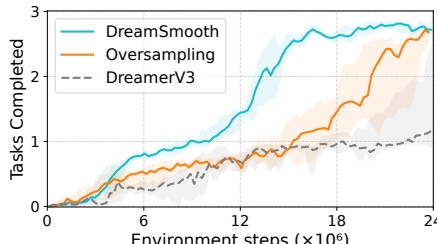

Figure 10: Oversampling sparse rewards ($p = 0.5$) improves DreamerV3 on RoboDesk, but still performs worse than DreamSmooth-Gaussian. The lines show median over 3 seeds, while shaded regions show maximum and minimum.

**Reward Model Size.** Another hypothesis for poor reward predictions is that the reward model does not have enough capacity to capture sparse rewards. To test this hypothesis, we increase the size of the reward model from $4$ layers of $768$ units, which DreamSmooth uses, to $5$ layers of $1024$ units and $6$ layers of $1280$ units, while keeping the rest of the world model the same. We observe in Figure 11 that without smoothing, increasing reward model size has negligible impact, and DreamSmooth outperforms all the reward model sizes tested. This indicates that the reward prediction problem is not simply caused by insufficient model capacity.

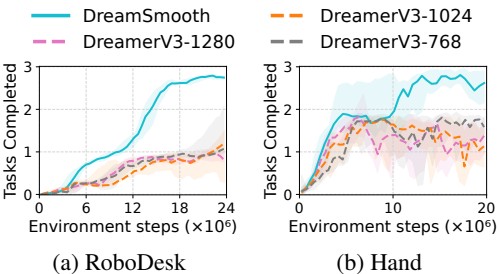

(a) RoboDesk          (b) Hand

Figure 11: Simply increasing the reward model size has negligible impact on performance. DreamerV3-768, 1024, and 1280 use $4$, $5$, $6$ layers of $768$, $1024$, $1280$ units, respectively.

**Loss Functions.** We verify whether other formulations of reward regression could solve the reward prediction problem in RoboDesk. Following Hafner et al. (2023), we take symlog of the prediction target for stable training regardless of the scale of rewards. We find in Figure 12 that reward prediction remains a challenge when using common loss functions, such as L1 and L2, with L1 significantly degrading task performance on RoboDesk. On the

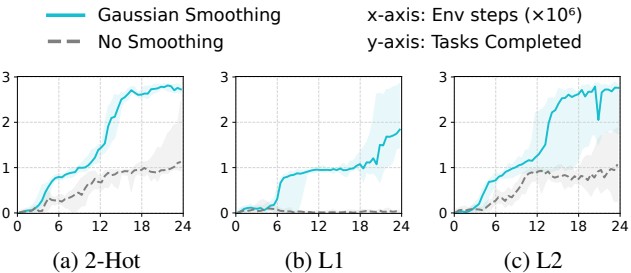

(a) 2-Hot          (b) L1          (c) L2

Figure 12: Learning curves (median over 3 seeds) of various loss functions for reward modeling on RoboDesk. Note that DreamerV3 and Dreamsmooth use the 2-Hot loss function.

other hand, applying reward smoothing improves performance for all three loss functions.

**Smoothing Parameter.** In Figure 13, we analyze the impact of the smoothing parameters $\sigma$ and $\alpha$ for Gaussian and EMA, respectively, on RoboDesk and Hand. We observe that DreamSmooth is insensitive to the smoothing parameters, performing well across a wide range of values.

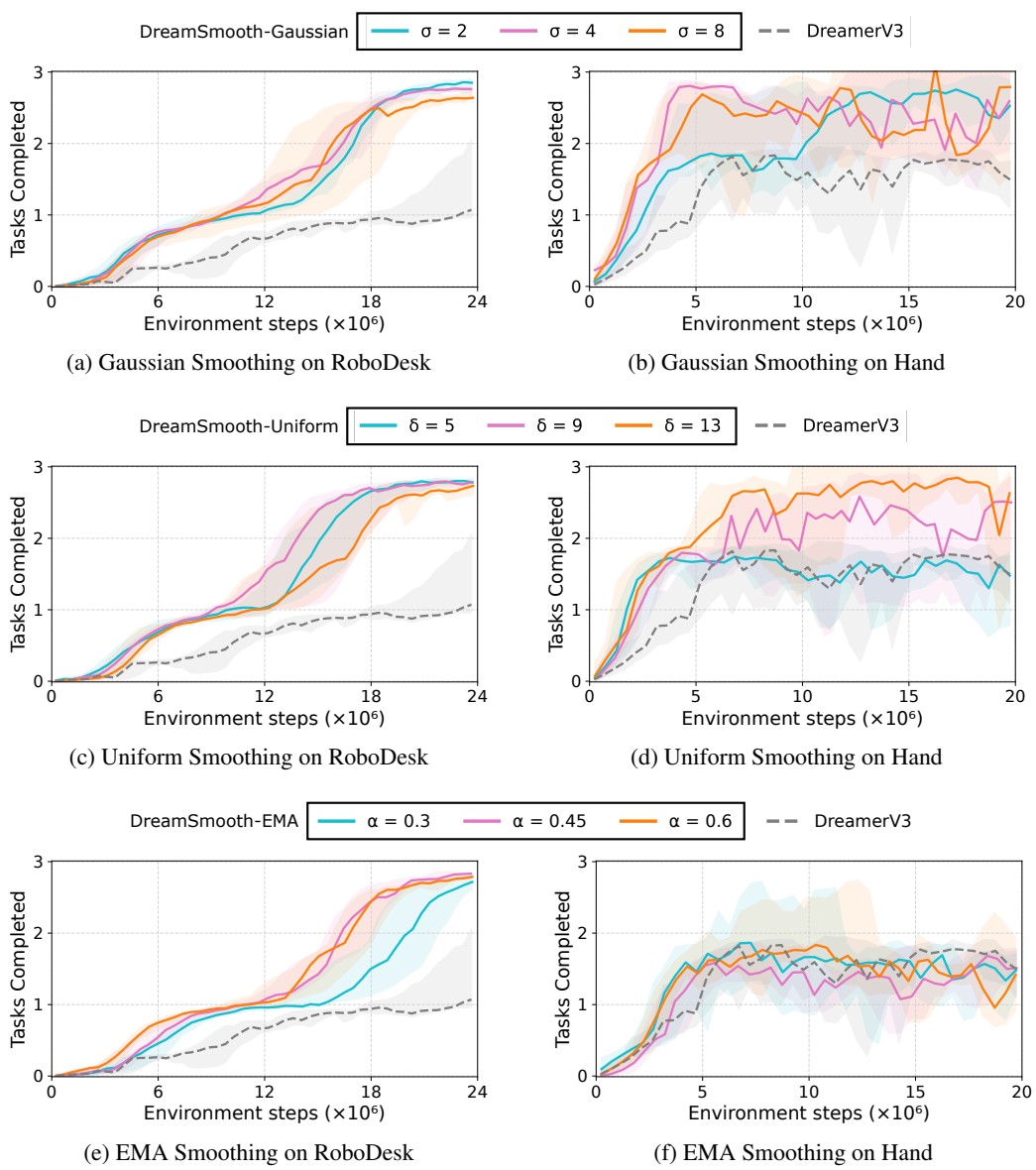

Figure 13: Parameter sweep over smoothing parameters $\sigma$, $\delta$, and $\alpha$. The lines show median task performance over 3 seeds, while shaded regions show maximum and minimum.

## 5 CONCLUSION

In this paper, we identify the reward prediction problem in MBRL and provide a simple yet effective solution: *reward smoothing*. Our approach, DreamSmooth, demonstrates superior performance in sparse reward tasks where reward prediction is not trivial mainly due to the partial observability or stochasticity of the environments. Moreover, DreamSmooth shows comparable results on the commonly used benchmarks, DMC and Atari, showing its task-agnostic nature. Although we show that our simple reward smoothing approach mitigates the difficulty in reward prediction, the improved reward prediction does not always improve the task performance, e.g., in Crafter. This can be because more predicted task rewards can also result in more false positives. Further investigation on this trade-off is a promising direction for future work.

ACKNOWLEDGMENTS

This work was supported in part by the BAIR Industrial Consortium, an ONR DURIP grant, Komatsu, and InnoHK Centre for Logistics Robotics. We would like to thank Seohong Park for proofreading our proof and all members of the Berkeley Robot Learning lab for their insightful feedback.

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

# A   PROOFS

Let $\mathcal{M} = (\mathcal{S}, \mathcal{A}, P, R, \gamma)$ be the given MDP. Without loss of generality, we assume the augmented form of the MDP $\mathcal{M}$, where a state $\boldsymbol{s}_t$ includes the entire history of states, i.e., $\boldsymbol{s}_t = (\boldsymbol{s}_1, \ldots, \boldsymbol{s}_t)$, and thus, reward functions $R, \tilde{R}$ have access to previous states, i.e., $\tilde{R}(\boldsymbol{s}_t) = \tilde{R}(\boldsymbol{s}_1, \ldots, \boldsymbol{s}_t)$.

**Theorem A.1.** *An optimal policy $\tilde{\pi}^*$ of the MDP with reward smoothing only with **past** rewards, e.g., EMA smoothing, $\tilde{\mathcal{M}} = (\mathcal{S}, \mathcal{A}, P, \tilde{R}, \gamma)$ is also optimal under the original MDP $\mathcal{M}$, where*

$$\tilde{R}(\boldsymbol{s}_t) = \sum_{i=-L}^{0} f_i \cdot \gamma^i R(\boldsymbol{s}_{t+i}) \quad and \quad \sum_{i=-L}^{0} f_i = 1. \tag{2}$$

*Proof.* We will use the theorem of reward shaping that guarantees an optimal policy introduced in Ng et al. (1999): if a modified reward function can be represented in the form of $R(\boldsymbol{s}_t) + \gamma\Phi(\boldsymbol{s}_{t+1}) - \Phi(\boldsymbol{s}_t)$ with any potential function $\Phi(\boldsymbol{s}_t)$, the new reward function yields the same optimal policy with the original reward function $R$.

Let the potential function for the EMA reward smoothing

$$\Phi(\boldsymbol{s}_t) = -\sum_{i=-L}^{-1} \gamma^i R(\boldsymbol{s}_{t+i}) + \sum_{i=-L}^{0} \gamma^i R(\boldsymbol{s}_{t+i}) \cdot \sum_{j=i+1}^{0} f_j. \tag{3}$$

Then, our reward shaping term in $\tilde{R}$ can be represented as the difference in the potential function $\gamma\Phi(\boldsymbol{s}_{t+1}) - \Phi(\boldsymbol{s}_t)$ as follows:

$$\gamma\Phi(\boldsymbol{s}_{t+1}) - \Phi(\boldsymbol{s}_t) = -R(\boldsymbol{s}_t) + \sum_{i=-L}^{0} f_i \cdot \gamma^i R(\boldsymbol{s}_{t+i}). \tag{4}$$

$$R(\boldsymbol{s}_t) + \gamma\Phi(\boldsymbol{s}_{t+1}) - \Phi(\boldsymbol{s}_t) = \sum_{i=-L}^{0} f_i \cdot \gamma^i R(\boldsymbol{s}_{t+i}) = \tilde{R}. \tag{5}$$

Hence, following Ng et al. (1999), reward shaping with our EMA smoothing guarantees the optimal policy in the original MDP $\mathcal{M}$. $\square$

However, Theorem A.1 does not apply to smoothing functions that require access to future rewards, e.g., Gaussian smoothing. As in Gaussian smoothing, a smoothed reward function may require future rewards, which are conditioned on the current policy; so is the reward model. In such cases, there is no theoretical guarantee; but in our experiments, we empirically show that reward models can adapt their predictions along the changes in policies and thus, improve MBRL.

Instead, we intuitively explain that an optimal policy under any reward smoothing (even though the reward function is post hoc and cannot be defined for MDPs) is also optimal under the original reward function.

**Theorem A.2.** *An optimal policy $\tilde{\pi}^*$ with the smoothed reward function $\tilde{R}$ is also optimal under the original reward function $R$, where*

$$\tilde{R}(\boldsymbol{s}_t) = \sum_{i=-L}^{L} \gamma^{clip(i,-t,T-t)} \cdot f_i \cdot R(\boldsymbol{s}_{clip(t+i,0,T)}) \quad and \quad \sum_{i=-L}^{L} f_i = 1. \tag{6}$$

*Proof.* First, we show that the discounted sum of original rewards $\sum_{t=0}^{T} \gamma^t R(\boldsymbol{s}_t)$ and the one of smoothed rewards $\sum_{t=0}^{T} \gamma^t \tilde{R}(\boldsymbol{s}_t)$ are the same for any trajectories $(\boldsymbol{s}_0, \boldsymbol{s}_1, \ldots, \boldsymbol{s}_T)$:

$$\sum_{t=0}^{T} \gamma^t \tilde{R}(\boldsymbol{s}_t) = \sum_{t=0}^{T} \gamma^t \sum_{i=-L}^{L} \gamma^{\text{clip}(i,-t,T-t)} \cdot f_i \cdot R(\boldsymbol{s}_{\text{clip}(t+i,0,T)}) \qquad \text{from Equation (6)} \quad (7)$$

$$= \sum_{t=0}^{T} \gamma^t R(\boldsymbol{s}_t) \cdot \sum_{i=-L}^{L} f_i \tag{8}$$

$$= \sum_{t=0}^{T} \gamma^t R(\boldsymbol{s}_t). \qquad \text{from } \sum_{i=-L}^{L} f_i = 1 \quad (9)$$

Let an optimal policy under the smoothed rewards $\tilde{R}$ be $\tilde{\pi}^*$. Assume that $\tilde{\pi}^*$ is not optimal under the original reward $R$. Then,

$$\exists \pi^*, \boldsymbol{s}_0 \quad \text{such that} \quad \mathbb{E}_{(\boldsymbol{s}_0,\ldots,\boldsymbol{s}_T) \sim \pi^*}\Big[\sum_{t=0}^{T} \gamma^t R(\boldsymbol{s}_t)\Big] > \mathbb{E}_{(\boldsymbol{s}_0,\ldots,\boldsymbol{s}_T) \sim \tilde{\pi}^*}\Big[\sum_{t=0}^{T} \gamma^t \tilde{R}(\boldsymbol{s}_t)\Big]. \tag{10}$$

However,

$$\mathbb{E}_{(\boldsymbol{s}_0,\ldots,\boldsymbol{s}_T) \sim \pi^*}\Big[\sum_{t=0}^{T} \gamma^t R(\boldsymbol{s}_t)\Big] = \mathbb{E}_{(\boldsymbol{s}_0,\ldots,\boldsymbol{s}_T) \sim \pi^*}\Big[\sum_{t=0}^{T} \gamma^t \tilde{R}(\boldsymbol{s}_t)\Big] \qquad \text{by Equation (9)} \quad (11)$$

$$> \mathbb{E}_{(\boldsymbol{s}_0,\ldots,\boldsymbol{s}_T) \sim \tilde{\pi}^*}\Big[\sum_{t=0}^{T} \gamma^t \tilde{R}(\boldsymbol{s}_t)\Big], \qquad \text{by Equation (10)} \quad (12)$$

which contradicts that $\tilde{\pi}^*$ is optimal under $\tilde{R}$. Therefore, the optimal policy $\tilde{\pi}^*$ under $\tilde{R}$ guarantees its optimality under $R$. $\qquad\square$

## B  Implementation Details

Models are trained on NVIDIA A5000, V100, RTX Titan, RTX 2080, and RTX 6000 GPUs. Each experiment takes about 72 hours for RoboDesk, 100 hours for Hand, 150 hours for Earthmoving, 96 hours for Crafter, and 6 hours for Atari and DMC tasks.

### B.1  Smoothing Functions in DreamSmooth

**Gaussian smoothing** follows the Gaussian distribution with $\sigma$:

$$f_i = k e^{\frac{-i^2}{2\sigma^2}}, \tag{13}$$

where $k = 1/(\sum_{i=-L}^{L} e^{\frac{-i^2}{2\sigma^2}})$ is a normalization constant.
We implement this using

```
scipy.ndimage.gaussian_filter1d(rewards, sigma, mode="nearest")
```

**Uniform smoothing** distributes rewards equally across $\delta$ consecutive timesteps.

$$f_i = \frac{1}{\delta} \quad \forall i \in \Big[-\frac{\delta-1}{2}, \frac{\delta-1}{2}\Big]. \tag{14}$$

We implement this using

```
scipy.ndimage.convolve(rewards, filter, mode="nearest")
```

**EMA smoothing** uses the following smoothing function:

$$f_i = \alpha(1-\alpha)^i \quad \forall i \leq 0, \tag{15}$$

which we implement by performing the following at each timestep:

```
reward[t] = alpha * reward[t - 1] + (1 - alpha) * reward[t]
```

## B.2 Model-based Reinforcement Learning Backbones

Hyperparameters for DreamerV3, TD-MPC, and MBPO experiments are shown in Table 1, Table 2, and Table 3, respectively.

Table 1: DreamerV3 hyperparameters. Episode length is measured in environment steps, which is the number of agent steps multiplied by action repeat. Model sizes are as listed in Hafner et al. (2023), which we also refer to for all other hyperparameters.

| Environment | Action Repeat | Episode Length | Train Ratio | Model Size | $\sigma$ | $\alpha$ | $\delta$ |
|---|---|---|---|---|---|---|---|
| Earthmoving | 4 | 2000 | 64 | L | 3 | 0.33 | 9 |
| RoboDesk | 8 | 2400 | 64 | L | 3 | 0.3 | 9 |
| Hand | 1 | 300 | 64 | L | 3 | 0.3 | 9 |
| Crafter | 1 | Variable | 64 | XL | 3 | 0.3 | 9 |
| DMC | 2 | 1000 | 512 | S | 3 | 0.33 | 9 |
| Atari | 4 | Variable | 1024 | S | 3 | 0.3 | 9 |

Table 2: TD-MPC hyperparameters. We refer to Hansen et al. (2022) for all other hyperparameters.

| Environment | Latent Dimension | CNN channels | Planning Iterations | $\sigma$ |
|---|---|---|---|---|
| Hand-Pixel | 128 | 64 | 6 | 3 |
| Hand-Proprio | 128 | – | 12 | 3 |

Table 3: MBPO hyperparameters. We refer to Janner et al. (2019) for all other hyperparameters.

| Environment | Layer Size | Prediction Head Size | Demo Pre-Training Steps | $\sigma$ |
|---|---|---|---|---|
| Hand-Proprio | 512 | 400 | 30000 | 3 |
| RoboDesk-Proprio | 512 | 400 | 30000 | 3 |
| DMC-Proprio | 256 | 200 | 0 | 3 |

## C Environment Details

### C.1 RoboDesk Environment

We use a modified version of RoboDesk (Kannan et al., 2021), where a sequence of manipulation tasks (`flat_block_in_bin`, `upright_block_off_table`, `push_green`) need to be completed in order. Figure 14 shows images of an agent successfully completing each of these tasks.

In the original environment, dense rewards are based on Euclidean distances of objects to their targets, with additional terms to encourage the arm to reach the object. They typically range from 0 to 10 per timestep. We use these dense rewards together with a large sparse reward of 300 for each task completed.

### C.2 Hand Environment

We modified the Shadow Hand environment (Plappert et al., 2018), so that the agent is required to achieve a sequence of pre-defined goal orientations in order. The first 3 goals are shown in Figure 15, while the subsequent goals are a repeat of the first 3. The goal orientations are chosen so that the agent only has to rotate the cube along the z-axis, and we only require the agent to match the cube's rotation to the goal, not its position.

In the original environment, dense rewards are computed using $r = -(10x + \Delta\theta)$, where $x$ is the Euclidean distance to some fixed position, and $\Delta\theta$ is the angular difference to the target orientation. In addition to these dense rewards, we provide a large sparse reward of 300 for each goal successfully achieved by the agent.

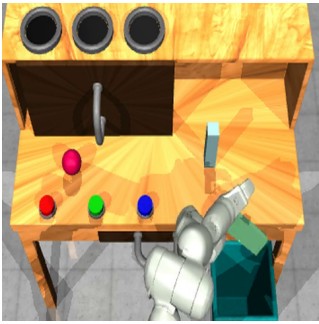
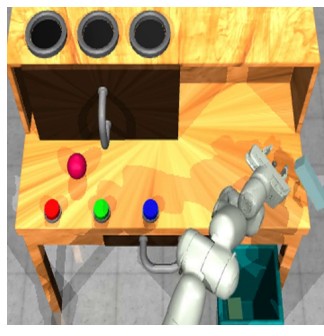
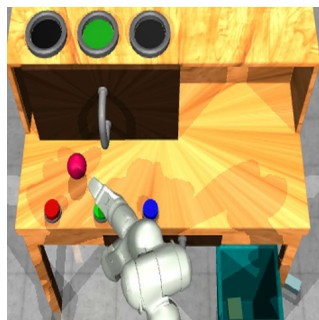

(a) Push green block into the bin     (b) Push teal block off the table     (c) Press the green button

Figure 14: Subtasks for RoboDesk.

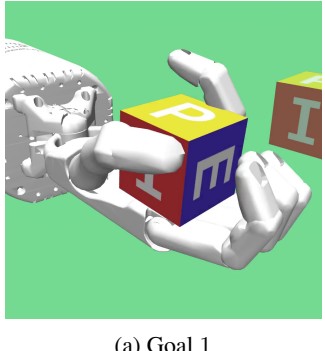

(a) Goal 1        (b) Goal 2        (c) Goal 3

Figure 15: Subtasks for Hand.

## C.3 AGX EARTHMOVING ENVIRONMENT

The Earthmoving environment consists of a wheel loader, dump truck, a pile of dirt, with some rocks on top of the pile. The environment is simulated using the realistic AGX Dynamics physics engine (Algoryx, 2020). The agent controls the wheel loader to pick up rocks and dump them in the dump truck.

The starting positions of the dirt pile, wheel loader, and dump truck are all randomized, as are the initial orientations of the dirt pile and wheel loader.

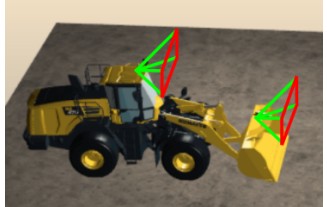

Figure 16: The agent uses one camera mounted on the cabin (left) for navigation, and one mounted on the bucket (right) for observing interactions with rocks and terrain.

The agent's observations consist of 3 components: a wide-angle egocentric RGB camera mounted on the cabin to allow navigation, an RGB camera mounted on the bucket for observing interactions with rocks, and proprioceptive observations (positions, velocity, speed, force of actuators etc.). We use $64 \times 64 \times 3$ images for all cameras, while the proprioceptive observation has 21 dimensions.

The action space is 4-dimensional: 2 dimensions for driving and steering the loader, and 2 dimensions for moving and tilting the bucket.

The reward consists of a large sparse reward for rocks picked up and dumped, and dense rewards for moving rocks towards the dumptruck. The total reward $r^t$ at timestep $t$ is computed using Equation (16).

$$r^t = \underbrace{\lambda_{\text{dump}}(m^t_{\text{dump}} - m^{t-1}_{\text{dump}}) + \lambda_{\text{load}}(m^t_{\text{load}} - m^{t-1}_{\text{load}})}_{\text{sparse reward}} + \underbrace{\lambda_{\text{move}} m^t_{\text{load}}(\max(2, d^t) - \max(2, d^{t-1}))}_{\text{dense reward}}$$

(16)

Where $m_{\text{dump}}$, $m_{\text{load}}$ are rock masses in the dumptruck and the bucket respectively, $d$ is the distance between the shovel and a point above the dumptruck, and $\lambda$ are constants.

# D DMC AND ATARI BENCHMARKING RESULTS

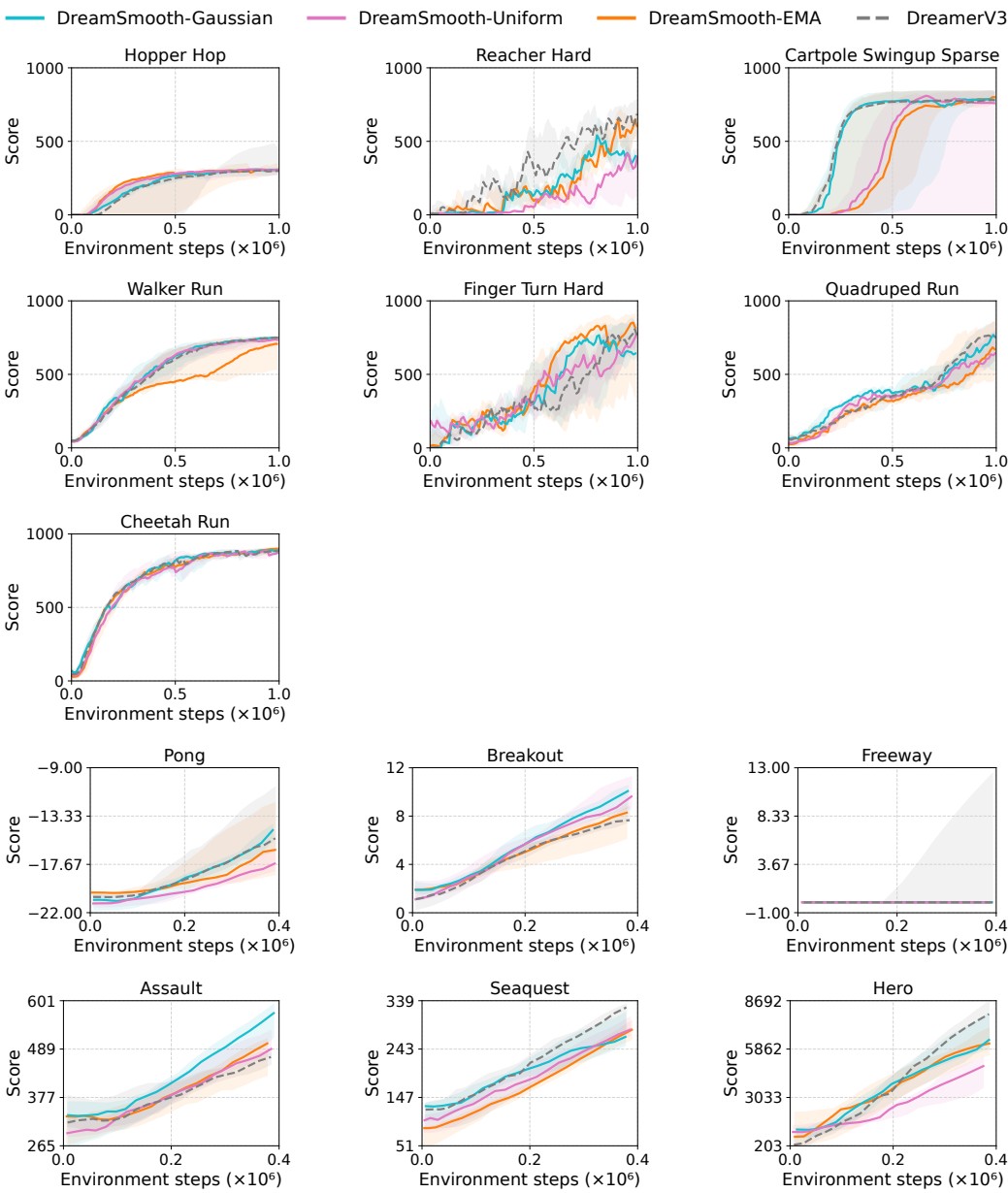

Figure 17: Full learning curves for the DMC and Atari benchmarks.

# E  MBPO

In addition to DreamerV3 and TD-MPC, we show the general applicability of our reward smoothing method with other MBRL algorithms, such as Model-Based Policy Optimization (MBPO) (Janner et al., 2019).

Similar to the experiments on DreamerV3, Figure 18 demonstrates that reward smoothing does not affect the performance of MBPO on the *dense-reward* DMC tasks (Walker Walk, Cheetah Run, Reacher Easy, Cartpole Swingup), while significantly improving the performance of MBPO on *sparse-reward* tasks, especially in the Hand environment. Since MBPO, when trained from scratch, struggles at solving Hand and RoboDesk, we initialize the replay buffer for these two experiments using trajectories from the fully-trained DreamerV3 policies (100 and 68 episodes, respectively). MBPO is then able to learn the first task in Hand with reward smoothing, while failing at both Hand and RoboDesk without reward smoothing. This indicates that DreamSmooth can also benefit Markovian models that have no access to neither past nor future states.

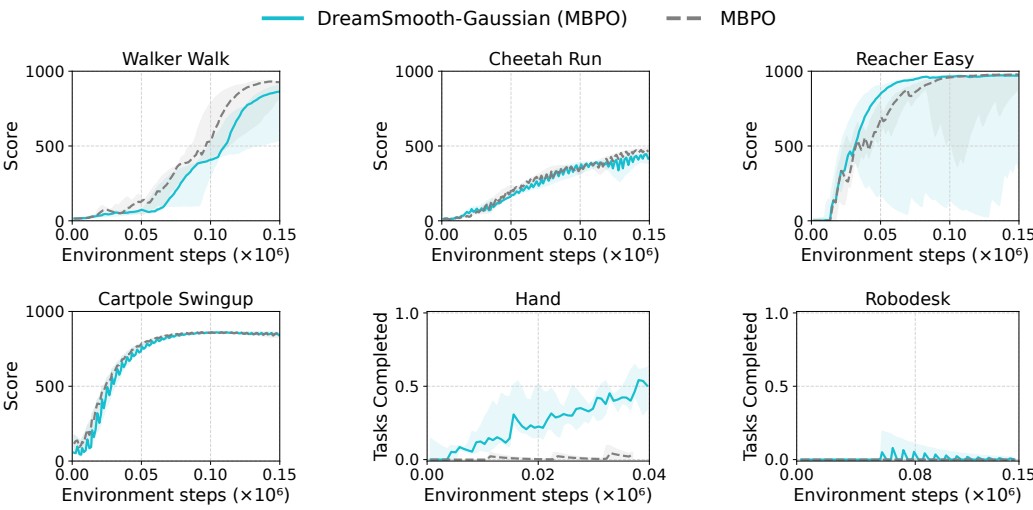

Figure 18: Learning curves for MBPO with DreamSmooth (Gaussian smoothing) on Hand, RoboDesk, and DMC tasks. For Hand and RoboDesk, MBPO trained from scratch could not achieve any meaningful reward signal. To ease the exploration problem, we initialize the replay buffers with demonstrations collected from the fully-trained DreamerV3 policies. The shaded regions show the minimum and maximum over 3 seeds.

## F    CRAFTER

It is surprising that despite improved reward predictions when using Gaussian or uniform smoothing on the Crafter environment, the task performance significantly deteriorates compared to vanilla DreamerV3, or even DreamerV3 with EMA reward smoothing, as can be seen in Figure 8d.

One possible cause is the symmetrical structure of the Gaussian and Uniform kernels, which makes the smoothed rewards dependent on both past and future ground truth rewards. This means that the reward model has to anticipate future rewards when performing predictions. We suspect that this leads to a high rate of **false positives** in Crafter, where there are many sources of sparse rewards. The false positives can make policy learning difficult.

To test this, we introduce 2 asymmetric variants of the uniform smoothing kernel:

Uniform $[-\delta, 0]$, where the smoothed rewards only depend on past ground truth rewards.

$$f_i = \frac{1}{\delta + 1} \quad \forall i \in [-\delta, 0]. \tag{17}$$

Uniform $[0, \delta]$, where the smoothed rewards only depend on future ground truth rewards.

$$f_i = \frac{1}{\delta + 1} \quad \forall i \in [0, \delta]. \tag{18}$$

The plots of predicted and ground truth rewards in Figure 20 and Figure 21 show that the reward models trained with the symmetric uniform smoothing kernel and Uniform $[0, 4]$ smoothing kernel have more false positives than those trained with Uniform $[-4, 0]$ and no smoothing, predicting significant positive rewards even when ground truth reward is $0$.

In Figure 19, smoothing kernels that require the model to anticipate future rewards perform worse than those that do not: Uniform $[-4, 0]$ performs the best, followed by symmetric uniform smoothing, then Uniform $[0, 4]$.

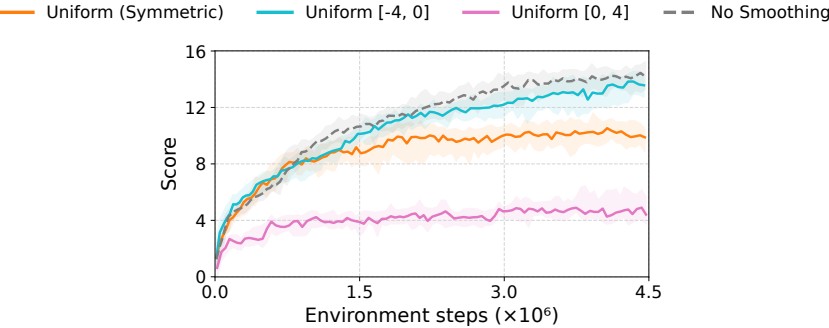

Figure 19: Learning curves of DreamSmooth with uniform smoothing variants on Crafter. Performance on Crafter is highly correlated with the need to anticipate future reward.

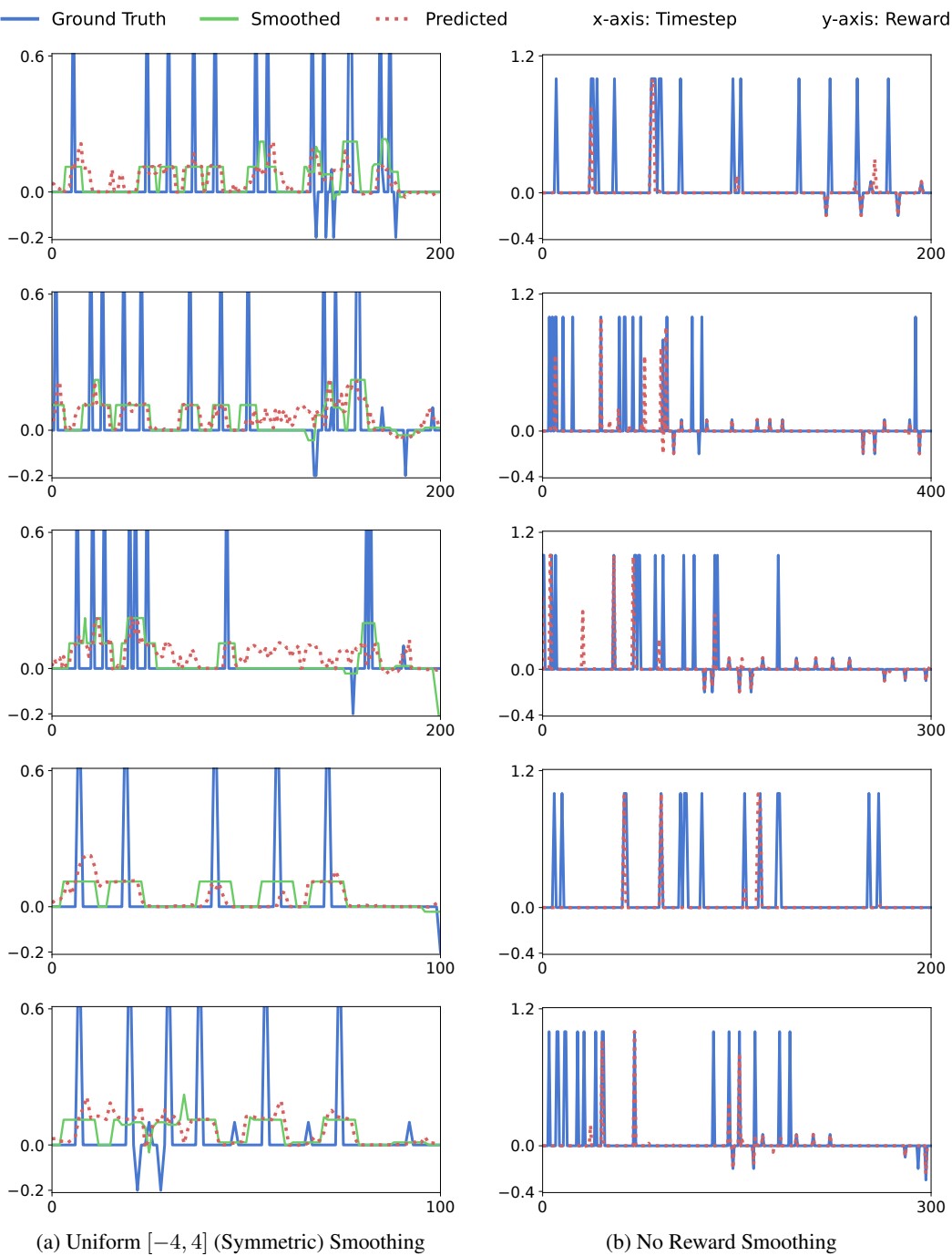

(a) Uniform $[-4, 4]$ (Symmetric) Smoothing          (b) No Reward Smoothing

Figure 20: Ground truth rewards and predicted rewards over five evaluation episodes on Crafter. Similar to Uniform $[0, 4]$ smoothing, the reward model trained with symmetric uniform smoothing predicts many false positives. Meanwhile, the reward model trained without smoothing predicts few false positives but has many false negatives.

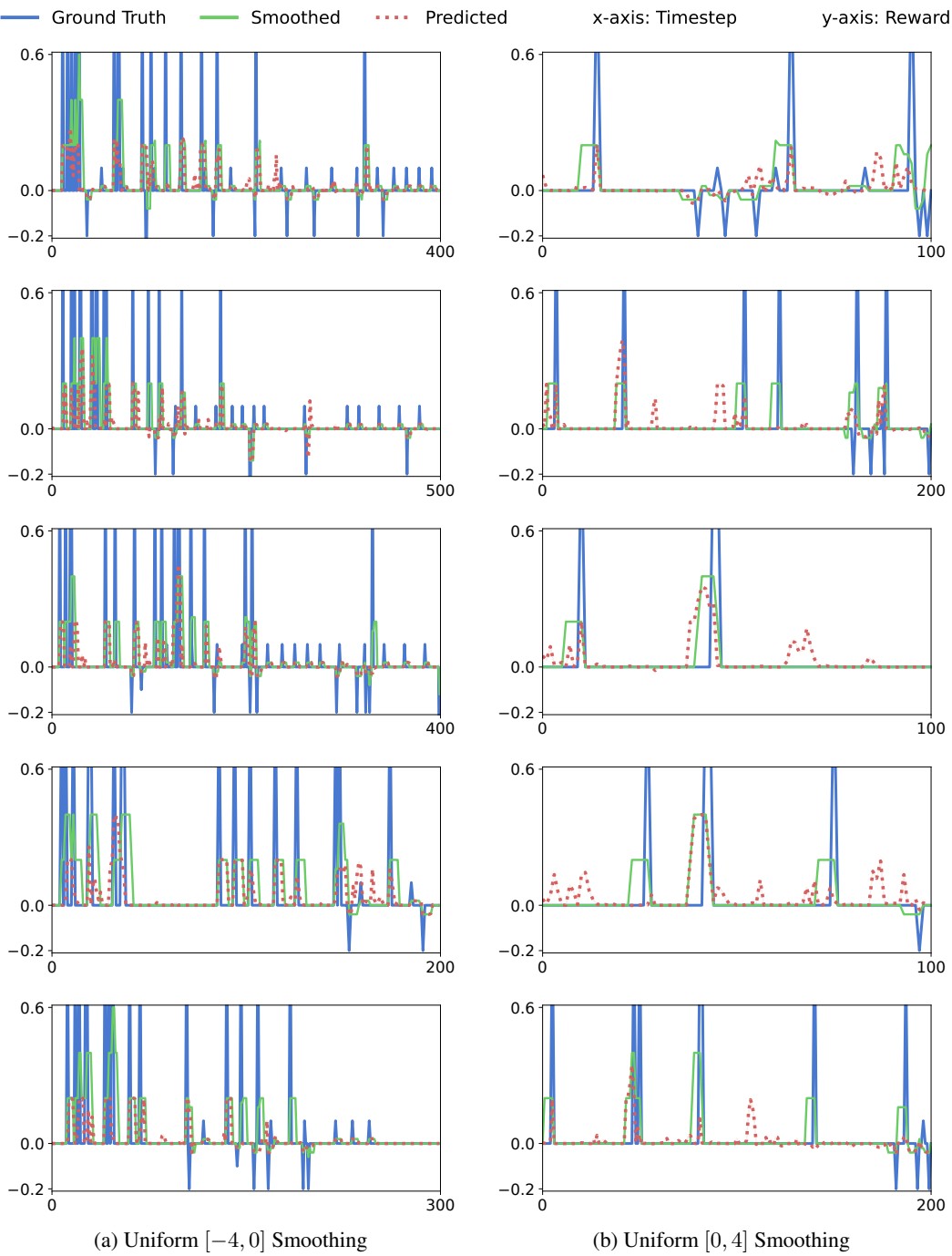

(a) Uniform $[-4, 0]$ Smoothing

(b) Uniform $[0, 4]$ Smoothing

Figure 21: Ground truth rewards and predicted rewards over five evaluation episodes on Crafter. With Uniform $[0, 4]$ smoothing, the reward model has to anticipate future rewards, resulting in more false positives than Uniform $[-4, 0]$, where the smoothed rewards at the current timestep only depend on past rewards.

