# OpenReview forum: "DreamSmooth: Improving Model-based Reinforcement Learning via Reward Smoothing"
_ICLR.cc/2024/Conference — ICLR 2024 poster_

### Official Review · Reviewer_Q6F6 · 2023-10-27

**Soundness:** 3 good
**Presentation:** 3 good
**Contribution:** 2 fair
**Rating:** 6
**Confidence:** 4

**Summary:**

This paper propose DreamSmooth which is based on Dreamer-V3. The technique used in this paper is to smooth the ground truth rewards when trainining Dreamer-V3. Experiments show that this modification works well on both dense and sparse rewards environments.

**Strengths:**

The proposed method solves a problem in model based RL when the gt rewards are sparse in these environments. Smoothing the rewards would make the reward function prediction process (reward leanring in model based RL) much better than before. At the same time, DreamSmooth shows it also performs well on dense rewards environments. It makes the whole algorithm more convincing. I think it is a good paper to investigate the reward smoothing technique for model based RL.

**Weaknesses:**

Dreamer-V3 has a symlog prediction function with reward learning process. I think different reward prediction function would contribute to the reward learning process. Do the authors conduct some experiments with different prediction function head to justify whether it could solve the sparse rewards problem? I think the issue discussed in this paper is mainly about the reward generalizability problem. It is hard for reward function in MBRL to generalize in sparse reward setting.

**Questions:**

I am curious about why DreamSmooth works comparable without reward smooth technique in dense reward environments. The reward smoothing technique changes the  reward distribution. As far as I am concerned, using this kind of technique would decline the final performance. Since MBRL like Dreamer is so important to this community, I lean to weak accept for this paper. However, I do have some concerns about this simple yet effective algorithms, especially why the final performance doesn't decline (especially on dense reward environments).

---

> ### Author Response · Authors · 2023-11-20
> **Author response**
>
> We appreciate the reviewer’s thorough and constructive feedback about our paper. We provide detailed answers to the questions below.
>
> &nbsp;
>
> **[Q] Do the authors conduct some experiments with different prediction function head to justify whether it could solve the sparse rewards problem?**
>
> This is a really good point! As suggested by the reviewer, we conducted additional experiments to see whether different reward head type (regression) and loss (L1 and L2) can solve the reward prediction problem.
>
> In our experiments on RoboDesk and Hand, both L1 and L2 losses with a typical MLP reward head miss many sparse reward signals, similar to the results with DreamerV3’s two-hot categorical regression. Furthermore, these reward modeling approaches also showed improved performances with DreamSmooth. We summarized these results in the updated paper (Figure 17, Appendix E).
>
> In summary, we found that the reward prediction problem exists in diverse commonly used regression models and losses, and DreamSmooth effectively alleviates this problem.
>
> &nbsp;
>
> **[Q] why DreamSmooth works comparable without reward smooth technique in dense reward environments.**
>
> We would like to refer the reviewer to Theorem A.1 in Appendix A, which shows that the proposed EMA reward smoothing guarantees an optimal policy even with the changes in reward distribution. Thus, smoothing the past rewards does not deteriorate the performance of MBRL.
>
> In the case when reward smoothing involves future rewards, Theorem A.2 explains that an optimal policy under any of our reward smoothing functions is also optimal under the original reward function. This still does not guarantee that MBRL can achieve optimal performance with reward smoothing; but, in practice, blurring reward signals a few steps before and after does not completely change learning dynamics, especially under the dense reward setting.
>
> &nbsp;
>
> ---
>
> &nbsp;
>
> We thank the reviewer again for the time and effort put into improving our paper. Please feel free to let us know if there are any additional concerns or questions.

---

> > ### Comment · Reviewer_Q6F6 · 2023-11-23
> > **Thanks for your response**
> >
> > Thanks for your response. I would like to keep my score.

---

### Official Review · Reviewer_7VX8 · 2023-10-31

**Soundness:** 3 good
**Presentation:** 3 good
**Contribution:** 3 good
**Rating:** 6
**Confidence:** 3

**Summary:**

The paper studies the challenges of learning reward models in the context of Model-Based Reinforcement Learning (MBRL). The authors argue that existing methods in MBRL fail to learn a good reward model in sparse / partially observable/stochastic environments/tasks and show empirical evidence for this. Based on the intuitive idea that, in such challenging scenarios, one only has to rely on rough reward estimates as humans do, they propose a method called DreamSmooth. In DreamSmooth the reward model is now tasked to predict a temporally smoothed reward instead of the exact reward. The authors propose 3 reward-smoothing schemes and empirically show that the approach can significantly improve the performance on most sparse reward scenarios.

**Strengths:**

These are the strengths of the paper in my opinion:

1) Studies a largely ignored, yet important problem of reward modelling in the context of MBRL.
2) Propose a simple yet effective solution for the same.
3) Well Written.

**Weaknesses:**

The major weaknesses are as follows:

1) Counter-intuitive results in Crafter, where the method performs worse even after having a much better reward model.
2) The need to experiment with 3 different reward smoothing schemes each with its own hyperparameters (since none of them seems to consistent favourite across tasks).

**Questions:**

1) Have you experimented with different loss functions (on the unsmoothed rewards)? For example, what would happen if you use an L1 Loss instead of an L2 Loss commonly used in literature?
2) The rationale behind why the counter-intuitive results in crafter is not convincing. Did the authors perform further empirical studies / analysis ?

---

> ### Author Response · Authors · 2023-11-20
> **Author response**
>
> We appreciate the reviewer’s thorough and constructive feedback about our paper. We provide detailed answers to the questions below.
>
> &nbsp;
>
> **[Q] Have you experimented with different loss functions (on the unsmoothed rewards)?**
>
> This is a really good point! As suggested by the reviewer, we conducted additional experiments to see whether different reward head type (regression) and loss (L1 and L2) can solve the reward prediction problem.
>
> In our experiments on RoboDesk, both L1 and L2 losses with a typical MLP reward head miss many sparse reward signals, similar to the results with DreamerV3’s two-hot categorical regression. Furthermore, these reward modeling approaches also showed improved performances with DreamSmooth. We summarized these results in the updated paper (Figure 17, Appendix E).
>
> In summary, we found that the reward prediction problem exists in diverse commonly used regression models and losses, and DreamSmooth effectively alleviates this problem.
>
>
> &nbsp;
>
> **[Q] The need to experiment with 3 different reward smoothing schemes each with its own hyperparameters**
>
> Thank you for pointing this out. We agree that it could be more convincing with the use of consistent hyperparameters across tasks.
>
> We have updated the results (Figure 8) in our experiments with the same hyperparameters across all tasks and the hyperparameters can be found in Table 1: $\sigma=3$ for Gaussian smoothing, $\alpha=0.3, 0.33$ for EMA smoothing, and $\delta=9$ for Uniform smoothing. Our reward smoothing methods show consistent results across tasks and we could observe that the effect of the smoothing hyperparameters are marginal.
>
> &nbsp;
>
> **[Q] The rationale behind why the counter-intuitive results in crafter is not convincing.**
>
> We have extended our analysis on Crafter in Appendix G. After carefully inspecting reward prediction of unsmoothed and smoothed rewards on Crafter, we found that Gaussian and uniform smoothing results in **more false positives** while achieving better prediction accuracy, as shown in Figure 20. We suspect this could be due to predicting rewards for the unknown future states. Asymmetric variants of the uniform smoothing kernel (Uniform [-4, 0] and Uniform [0, 4]) in Figure 21 clearly show that reward smoothing kernels dependent on future rewards are prone to predicting positive rewards even when the ground truth reward is 0.
>
> DreamSmooth’s high rate of false positives can be one reason for the poor performance on Crafter despite its high precision and recall on reward prediction. It is still surprising how DreamerV3 could perform well while missing most of the reward predictions. We leave this investigation as future work.
>
> &nbsp;
>
> ---
>
> &nbsp;
>
> We thank the reviewer again for the time and effort put into improving our paper. Please feel free to let us know if there are any additional concerns or questions.

---

> > ### Comment · Reviewer_7VX8 · 2023-11-21
> >
> > Thank you for your reply and further analysis!! The reviewers have addressed most of my questions and concerns... I maintain my postiive opinion of the paper!

---

### Official Review · Reviewer_BjnM · 2023-11-01

**Soundness:** 3 good
**Presentation:** 3 good
**Contribution:** 2 fair
**Rating:** 5
**Confidence:** 4

**Summary:**

This paper introduces DreamSmooth, a simple and effective method that improves the performance of model-based RL on sparse reward environments. The authors observe that on sparse reward tasks, it is challenging to fit an accurate reward model due to data imbalance. This in turn bottlenecks the performance of model-based RL methods like Dreamer. To mitigate this problem, the authors apply a smoothing function to the reward, effectively spreading the reward signal to adjacent states in the trajectory. The authors propose three reward smoothing functions: Gaussian, Uniform, and EMA. While EMA is the only function that guarantees policy invariance, all three are empirically found to work well, resulting in more accurate reward predictions and higher performance in sparse reward tasks compared to baselines.

**Strengths:**

- Reward sparsity is a long-standing problem in model-based RL. Even with an accurate dynamics model, policy optimization would not work if the reward model fails to capture sparse reward signals.
- The method is extremely simple, with only a one-line change to the Dreamer code, yet it brings a significant improvement across a suite of challenging sparse reward tasks.
- The authors performed extensive analysis and ablation studies to identify the root cause of MBRL failure and demonstrate the effectiveness of their algorithm.

**Weaknesses:**

- While reward prediction error is indeed one consequence of reward sparsity, the fundamental challenge that comes with sparse rewards is exploration. If there is no reward signal in the first place, then reward smoothing does not work either. While this paper provides a simple remedy to alleviate reward sparsity, it does not address the fundamental exploration problem.
- Two of the smoothing functions are unable to guarantee policy invariance, and it is possible to construct adversarial examples (see questions).

**Questions:**

- It seems that reward smoothing can potentially lead to reward ambiguity. For example, if a bad state is visited right after a successful state vs. after a sequence of bad states, it would get assigned different smoothed reward values. How does reward ambiguity affect MBRL methods? I suspect the recurrent architecture of Dreamer helps mitigate this issue. To verify, can you run a Markovian MBRL method like MBPO with reward smoothing and see if there's any improvement there?
- There are inductive biases built into each smoothing function. For example, Gaussian and Uniform smoothing functions assume symmetry. However, this may not align with the environment dynamics. Consider a ball rolling off a staircase and receiving a sparse reward right at the edge of the staircase. The smoothing function bumps up the reward of the states before and after falling off the staircase, but in practice, it is much harder to climb back up from the lower platform than to roll down. In other words, reward smoothing can give rise to overoptimistic behaviors. Do you see this reflected in any task?
- Does reward smoothing benefit model-free methods such as policy gradient?

---

> ### Author Response · Authors · 2023-11-20
> **Author response**
>
> We appreciate the reviewer’s thorough and constructive feedback about our paper. We provide detailed answers to the questions below.
>
> &nbsp;
>
> **[Q] the fundamental challenge that comes with sparse rewards is exploration. … it does not address the fundamental exploration problem.**
>
> As the reviewer pointed out, our paper is **not** tackling all the challenges in solving sparse-reward tasks. Instead, we try to bring attention to one of the challenges in sparse-reward tasks, **the reward prediction problem** in model-based RL, which can happen even when exploration is not very difficult. For example, vanilla DreamerV3 could get sparse rewards in all our experiments but fails to correctly predict these sparse rewards and ends up with poor final performance. We made this point clearer in the abstract and introduction of the revised paper by removing some “sparse reward”.
>
> &nbsp;
>
>
> **[Q] reward smoothing can give rise to overoptimistic behaviors. Do you see this reflected in any task?**
>
> Our reward smoothing will assign some positive reward to a bad state after a good state. This is not a big problem because of the following reasons.
>
> * After reward smoothing, the bad state (A) after the good state (B) can be assigned with some positive reward. But, if there is a better state (C) after (B) then (C) will get a larger reward than (A) after reward smoothing. Thus, an RL policy will learn to move toward the state with relatively larger original rewards, which will be also larger after smoothing.
>
> * As the reviewer mentioned, the recurrent state space model used in Dreamer helps discern bad states followed after good states. In the staircase example, the rewards will be bumped up right after falling off the staircase but this reward doesn’t last long since the past history tells an agent that it is far away from a rewarding state.
>
> Many of our experiments reflect this property: once an agent gets a sparse reward, coming back to the same state does not give additional rewards.
>
> &nbsp;
>
>
> **[Q] How does reward ambiguity affect MBRL methods? … can you run a Markovian MBRL method like MBPO with reward smoothing and see if there's any improvement there?**
>
> Thank you for an interesting suggestion! We conducted additional experiments with a Markovian model-based RL method, MBPO [1], on the state-based environments, Hand and RoboDesk. Unfortunately, MBPO failed to learn any of these tasks, so we could not verify whether reward smoothing improves the performance or not. We will further investigate this in an easier environment in the camera-ready version.
>
> On the dense-reward DMC tasks, DreamSmooth does not hurt the Markovian MBRL method, MBPO, as shown in Appendix, Section F and Figure 18.
>
> &nbsp;
>
>
> **[Q] Does reward smoothing benefit model-free methods?**
>
> The benefit of our reward smoothing approach is easier reward model learning. Thus, our approach does not help model-free methods, which do not learn to predict rewards.
>
> &nbsp;
>
>
> ---
> &nbsp;
>
>
> We hope our response addresses all your concerns and questions. Please let us know if there are any additional concerns or questions.
>
> &nbsp;
>
>
>
> [1] Janner et al. “When to Trust Your Model: Model-Based Policy Optimization” NeurIPS 2019

---

> > ### Comment · Reviewer_BjnM · 2023-11-20
> >
> > Thanks for addressing my comments. I have the feeling that the effectiveness of reward smoothing hinges on the recurrent structure of Dreamer. This feels like overfitting to the Dreamer line of work and can be included in e.g. Dreamer V4. I would really like to see the broader impact that reward smoothing has, for example, on model-based RL with Markovian models.

---

> > > ### Author Response · Authors · 2023-11-23
> > >
> > > As per Reviewer BjnM, we got new positive results with Markovian models (MBPO) in Appendix F and Figure 18 of the revised paper. In summary, DreamSmooth also benefits MBPO on the sparse-reward Hand task even though MBPO uses Markovian models. Together with the improvements of TD-MPC+DreamSmooth shown in Figure 9, this result clearly shows that DreamSmooth is generally applicable in model-based RL in practice.
> > >
> > > Thank you so much for your valuable feedback and we hope our new results address your concerns.

---

### Official Review · Reviewer_4wgT · 2023-11-01

**Soundness:** 3 good
**Presentation:** 3 good
**Contribution:** 2 fair
**Rating:** 6
**Confidence:** 4

**Summary:**

In model-based reinforcement learning (MBRL), it is crucial to correctly estimate the reward model. However, when the rewards in the environment are sparse, it poses challenges in learning the reward function. The authors have shown convincing examples that the algorithm may achieve a smaller loss by simply predicting zero rewards, than predicing the sparse rewards at an incorrect time step.

This work remedies this problem by asking the algorithm to learn a smooth reward function. The proposed algorithm is based on the DreamerV3 algorithm and evaluated on a wide range of tasks.

**Strengths:**

This work proposes a simple yet effective approach to improve an MBRL agent’s performance in environments with sparse rewards. The algorithm is evaluated on simulated robotic control, 2D navigation, and Atari game domains. The authors also conducted ablation studies to show that this approach outperforms some other baseline algorithms to address the sparse reward issue, including oversampling sequences with sparse rewards, increasing reward model size, etc.

Additionally, this work also empirically verifies the challenges of reward learning in MBRL, finding out that the agent may achieve a smaller loss by predicting zero rewards than by predicting wrong rewards.

**Weaknesses:**

**Novelty.** This is more like an engineering trick that the community has considered as an ad-hoc approach to resolve to learn in environments with sparse rewards. Although this smoothing technique intuitively makes sense, I didn’t see justifications for the correctness of this approach. See Question 1 below.

**Baseline method.** DreamV3 is the only baseline method for almost all the tasks, except that TD-MPC is used for the Hand Task. Unless this work only considers robotic tasks, other more popular RL algorithms need to be included. Also, if this work is indeed only constrained to robotic tasks, I believe the authors need to make that clear in the paper, and also explain why this simple technique cannot be applied to other RL algorithms.

**Questions:**

1. When we change the reward function, is the agent’s policy guaranteed to have a high value under the original reward function?
When sparse rewards indeed specify critical states that have high rewards, would a smooth reward function blur out the true critical states, so that the optimal policy does not visit the critical states?

2. Is there any rationale for using DreamV3 as the baseline for most tasks?

---

My questions and concerns about weaknesses are addressed in the rebuttal.

---

> ### Author Response · Authors · 2023-11-20
> **Author response**
>
> We appreciate the reviewer’s thorough and constructive feedback about our paper. We provide detailed answers to the questions below.
>
> &nbsp;
>
> **[Q] This is more like an engineering trick…**
>
> We would like to highlight that the major contribution of this paper is to bring attention to reward modeling in MBRL, which has been largely overlooked and not actively discussed in the field. We provide empirical evidence that shows the importance of accurate reward modeling and the effectiveness of our simple approach.
>
> In deep learning, many engineering tricks have been playing a critical role despite their simplicity. We believe it’s worth discussing such engineering tricks, which suggests a new perspective or problem to the community; and we believe our paper does the same.
>
> &nbsp;
>
> **[Q] Is there any rationale for using DreamerV3 as the baseline for most tasks?**
>
> We mainly used DreamerV3 since it is the **only** model-based RL method that consistently works on a variety of tasks with high-dimensional inputs, sparse rewards, and long episode lengths. In Figure 9 of the original submission, TD-MPC could learn the Hand task and TD-MPC w/ DreamSmooth significantly improved TD-MPC. But, TD-MPC failed to learn other sparse-reward tasks.
>
> We also conducted additional experiments with another model-based RL method, MBPO [1], on the state-based environments, Hand and RoboDesk. However, MBPO couldn’t learn any of these tasks. We included these results in Section F and Figure 18.
>
> &nbsp;
>
> **[Q] I didn’t see justifications for the correctness of this approach. … would a smooth reward function blur out the true critical states, so that the optimal policy does not visit the critical states?**
>
> We would like to refer the reviewer to Theorem A.1 in Appendix A, which shows that the proposed EMA reward smoothing guarantees an optimal policy. This requires a policy to access the history of states and the recurrent state structures used in many model-based RL methods enable a policy under smoothed rewards to hold Theorem A.1.
>
> In the case when reward smoothing involves future rewards, Theorem A.2 explains that an optimal policy under any of our reward smoothing functions is also optimal under the original reward function. This still does not guarantee that MBRL can achieve optimal performance with reward smoothing; but, in practice, blurring reward signals a few steps before and after does not completely change learning dynamics since a policy learns from a value function (or MC value estimate in TD-MPC), which already blurs out the rewards on the critical states.
>
> &nbsp;
>
> **[Q] if this work is indeed only constrained to robotic tasks, I believe the authors need to make that clear in the paper…**
>
> This work is *not* specifically designed for the robotic domain. Our paper includes the most popular RL benchmarks, Atari and DMC. Moreover, Crafter and Atari benchmarks in our experiments are not robotic tasks.
>
> We used many robotic environments since these tasks have challenging properties of partial observability and sparse reward, which makes reward prediction difficult. We are happy to try our method on other non-robotic RL benchmarks with similar properties.
>
> &nbsp;
>
> ---
>
> &nbsp;
>
> We hope our response addresses all your concerns and questions. Please let us know if there are any additional concerns or questions.
>
> &nbsp;
>
> [1] Janner et al. “When to Trust Your Model: Model-Based Policy Optimization” NeurIPS 2019

---

> > ### Comment · Reviewer_4wgT · 2023-11-20
> > **Thanks for your response**
> >
> > Thanks for your responses and the additional results. You have mostly addressed my concerns:
> >
> > * There are indeed theoretical justifications for EMA (based on reward shaping).
> > * I agree that "engineering tricks" are not necessarily negative, as long as there are theoretical justifications and being evaluated on a wide range of tasks.
> > * The evaluation domains include two tasks that are not robotic tasks -- Craft and Atari.
> >
> > I still have a question about Theorem A.1 and A.2. The authors claimed that "there is no theoretical guarantee" for smoothing functions that require access to future rewards. However, Theorem A.2 claims that a broader class of reward smoothing functions (expressed as Eq. 6), which can have access to future rewards, can also preserve optimal policies. What kind of theoretical guarantees does A.2 provide?

---

> > > ### Author Response · Authors · 2023-11-21
> > > **Author response #2**
> > >
> > > Thank you for your prompt response and we are happy to hear that our response addressed most of your concerns!
> > >
> > > As the reviewer noticed, Theorem A.2 shows that our reward smoothing preserves optimal policies. However, a new MDP after reward smoothing cannot be defined in a conventional way since a smoothed reward function now additionally depends on a policy (i.e. future states and rewards). Therefore, we cannot provide a convergence guarantee to the optimal value function and policy in the smooth-reward setting, unlike in the original reward setting.
> > >
> > > Even so, our experiments indicate that this is not an issue in practice: DreamerV3 and TD-MPC successfully learn policies even with reward smoothing using future rewards, showing significant improvements on sparse-reward tasks while maintaining similar performance in dense reward environments.
> > >
> > >
> > > Please let us know if you have any further questions or concerns!

---

> > > > ### Comment · Reviewer_4wgT · 2023-11-21
> > > > **Thanks for the clarifications**
> > > >
> > > > Thanks for the clarifications! Indeed, the new MDP after reward smoothing is not conventional as the reward of a state depends on the states before and after it in a trajectory. It may be helpful to understand the theoretical implications on optimal policy perservence in this kind of MDPs, which may not be within the scope of this paper though.
> > > >
> > > > I have increased my score.

---

### Meta-Review · Area_Chair_TUSU · 2023-12-06

**Metareview:**

DreamSmooth, addresses the challenge of learning a reward model from a sparse signal in the context of Model-Based Reinforcement Learning (MBRL). This is achieved by predicting a temporally smoothed reward instead of the exact reward. The paper studies three reward smoothing functions: Gaussian, Uniform, and EMA. While EMA is the only function that guarantees policy invariance, all three are empirically found to work well, resulting in more accurate reward predictions and higher performance in sparse reward tasks compared to baselines. The choice of smoothing functions is not very well explained, and theoretical policy convergence guarantees are typically lost by the application of DreamSmooth.

**Justification For Why Not Higher Score:**

The proposed method might be viewed as a minor modification of Dreamer-V3. The wide applicability of the approach has not been sufficiently justified, especially since applying a smoothing function is not always an optimal-policy preserving transformation and may break convergence guarantees.

**Justification For Why Not Lower Score:**

DreamSmooth shows significant improvement in the agent's performance. It is a simple and elegant mechanism to address the challenge faced by reward sparsity.

---

### Decision · Program_Chairs · 2024-01-16

Accept (poster)